# Association of lithocholic acid with skeletal muscle hypertrophy through TGR5-IGF-1 and skeletal muscle mass in cultured mouse myotubes, chronic liver disease rats and humans

**Yasuyuki Tamai[1†], Akiko Eguchi[1*†], Ryuta Shigefuku[1], Hiroshi Kitamura[2], Mina Tempaku[1], Ryosuke Sugimoto[1], Yoshinao Kobayashi[3], Motoh Iwasa[1], Yoshiyuki Takei[1], Hayato Nakagawa[1]**

[1]Department of Gastroenterology and Hepatology, Graduate school of medicine, Mie University, Tsu, Japan; [2]Department of Veterinary Medicine, School of Veterinary Medicine, Rakuno Gakuen University, Ebetsu, Japan; [3]Center for Physical and mental health, Mie University Graduate School of Medicine, Tsu, Japan

**\*For correspondence:**
akieguchi@med.mie-u.ac.jp

[†]These authors contributed equally to this work

**Competing interest:** The authors declare that no competing interests exist.

## Abstract

**Background:** Hepatic sarcopenia is one of many complications associated with chronic liver disease (CLD) and has a high mortality rate; however, the liver-muscle axis is not fully understood. Therefore, few effective treatments exist for hepatic sarcopenia, the best of which being branched-chain amino acid (BCAA) supplementation to help increase muscle mass. Our aim was to investigate the molecular mechanism(s) of hepatic sarcopenia focused on bile acid (BA) composition.

**Methods:** The correlation between serum BA levels and psoas muscle mass index (PMI) was examined in 73 CLD patients. Gastrocnemius muscle phenotype and serum BA levels were assessed in CLD rats treated with BCAA. Mouse skeletal muscle cells (C2C12) were incubated with lithocholic acid (LCA), G-protein-coupled receptor 5 (TGR5) agonist or TGR5 antagonist to assess skeletal muscle hypertrophy.

**Results:** In human CLD, serum LCA levels were the sole factor positively correlated with PMI and were significantly decreased in both the low muscle mass group and the deceased group. Serum LCA levels were also shown to predict patient survival. Gastrocnemius muscle weight significantly increased in CLD rats treated with BCAA via suppression of protein degradation pathways, coupled with a significant increase in serum LCA levels. LCA treated C2C12 hypertrophy occurred in a concentration-dependent manner linked with TGR5-Akt pathways based upon inhibition results via a TGR5 antagonist.

**Conclusions:** Our results indicate LCA-mediated skeletal muscle hypertrophy via activation of TGR5-IGF1-Akt signaling pathways. In addition, serum LCA levels were associated with skeletal muscle mass in cirrhotic rats, as well as CLD patients, and predicted overall patient survival.

**Funding:** This research was supported by JSPS KAKENHI Grant Number 22K08011 and 21H02892, and AMED under Grant Number JP21fk0210090 and JP22fk0210115. Maintaining cirrhotic rats were partially supported by Otsuka Pharmaceutical Company.

## Editor's evaluation

Alterations in skeletal muscle mass, in particular sarcopenia, are of central clinical importance. This paper examines a particular subclass of etiologies related to the chronic liver disease for this condition. This paper identifies a potential causative for this for the first time by using both in vivo and in vitro techniques to draw associations between bile acid concentrations and liver disease. The paper is of interest to both basic scientists and clinicians.

## Introduction

Sarcopenia is the loss of muscle mass, strength, and physical function. Sarcopenia is classified into two categories, primary being age-related and secondary encompassing other causes such as chronic diseases. Hepatic sarcopenia differs from aging sarcopenia insofar as it is defined by a rapid decrease in muscle mass and power. Hepatic sarcopenia is one in the panoply of complications associated with chronic liver diseases (CLDs), in particular, liver cirrhosis, with its high mortality (or low survival rate) and poor post-liver transplantation outcomes (*Ebadi et al., 2019*; *Hara et al., 2016*). A variety of factors are altered in hepatic sarcopenia, including decreased serum branched-chain amino acid (BCAA) levels (*Tajiri and Shimizu, 2018*), decreased serum vitamin D levels (*Okubo et al., 2020*), increased bile acids (BAs) (*Kobayashi et al., 2017*), abnormal insulin growth factor-1 (IGF-1) and mammalian target of rapamycin (mTOR) signaling (*Allen et al., 2021*), increased reactive oxygen species, and increased inflammatory cytokines and myostatin expression (*Ebadi et al., 2019*; *Allen et al., 2021*). BCAA supplementation has been shown to significantly improve skeletal muscle mass index measurements (*Ismaiel et al., 2022*). In contrast, anti-myostatin monoclonal neutralizing antibodies developed by several companies failed in clinical trials targeted to treat Duchenne muscular dystrophy (*Wagner, 2020*). The molecular mechanisms underpinning the muscle-liver axis involved in hepatic sarcopenia are not fully understood. Therefore, the elucidation of molecular mechanisms and effective treatment designs is required to prevent the progression of hepatic sarcopenia and to improve overall patient prognosis.

CLD has a major impact on BA composition (*Sauerbruch et al., 2021*). BAs are amphipathic steroid molecules synthesized from cholesterol and are categorized as being primary or secondary. Primary BAs are synthesized and conjugated in hepatocytes and secreted into the intestine. Most conjugated BAs undergo deconjugation and dehydration by intestinal bacteria, resulting in the production of secondary BAs. BA pools containing a mix of primary and secondary BAs are essential for solubilizing lipids and fat-soluble vitamins, thus promoting their absorption into the small intestine (*Arab et al., 2017*; *Guzior and Quinn, 2021*). In addition to their canonical function in digestion, BAs are known to act as signaling molecules that regulate metabolic pathways, such as glucose, lipid, and energy homeostasis, through various receptors including G-protein-coupled receptor 5 (TGR5), farnesoid X receptor, and vitamin D receptor (*Arab et al., 2017*). TGR5 activation induced by cholic acid (CA), chenodeoxycholic acid (CDCA), deoxycholic acid (DCA), and lithocholic acid (LCA) as part of the overall BA composition is a key event regulating skeletal muscle cells with the most potent endogenous ligand for TGR5 being LCA (*Pols et al., 2011*). Indeed, LCA, a secondary BA, induced TGR5 activation in skeletal muscle and enhanced muscle mass hypertrophy in mice through an increase in IGF-1, a known muscle hypertrophy-related gene (*Sasaki et al., 2018*). However, the role of LCA in cirrhotic liver disease-related sarcopenia has not been fully clarified.

In this study, we investigate the interaction between BAs, including LCA, and skeletal muscle mass, in CLD patients, as well as CLD rats, and explore the molecular mechanism of LCA on skeletal muscle hypertrophy in cultured mouse myotubes, C2C12.

## Methods

### Patients and serum BA measurements in human

The study protocol (H2019-063) was approved by the Clinical Research Ethics Review Committee of Mie University Hospital. This study was performed retrospectively on stored samples, and subjects were allowed to opt out of their data being used. Written informed consent was obtained from all subjects at the time of blood sampling. A total of 113 treatment-naïve patients with hepatocellular

carcinoma (HCC) hospitalized in the Department of Gastroenterology and Hepatology, Mie University Hospital for treatment of HCC between January 2015 and January 2017 were included as a retrospective study. HCC diagnosis was based on clinical history, serologic testing, and radiologic imaging. 36 patients were excluded due to oral administration of ursodeoxycholic acid. Three patients who had other malignancies within the past 3 years were excluded. One patient was excluded due to kidney transplant. As a result, a total of 73 patients with CLD were analyzed for the current study. Patients positive for HBsAg were diagnosed with HBV infection, whereas those positive for anti-HCV were diagnosed with HCV infection. Alcohol associated liver disease was defined as alcohol consumption >60 g/day. Nonalcoholic steatohepatitis (NASH) was diagnosed based on pathological findings and/or fatty liver without any other evident causes of CLDs (viral, autoimmune, genetic, etc.). Hepatic functional reserve was categorized by albumin-bilirubin (ALBI) score (*Johnson et al., 2015*). The psoas muscle mass index (PMI) (psoas muscle area at the middle of the third lumbar vertebra [L3] [$cm^2$]/height [$m^2$]) was manually calculated from CT images. All treatments were performed following the Japanese practical guidelines for HCC as possible (*Kokudo et al., 2019*). Post-HCC treatment follow-up consisted of laboratory tests, including tumor markers, every 3 months and dynamic CT or MRI every 6 months.

BA concentrations were determined in a blind as described by Ando et al. with minor modifications (*Murakami et al., 2018*; *Ando et al., 2006*). After the addition of internal standards and 0.5 M potassium phosphate buffer (pH 7.4), BAs were extracted with Bond Elut C18 cartridges and quantified by Liquid Chromatograph-Mass Spectrometry(LC-MS)/MS. Chromatographic separation was performed using a Hypersil GOLD column (150×2.1 mm, 3.0 μm; Thermo Fisher Scientific) at 40°C. The mobile phase consisted of (i) 20 mM ammonium acetate buffer (pH 7.5)-acetonitrile-methanol (70:15:15, v/v/v) and (ii) 20 mM ammonium acetate buffer (pH 7.5)-acetonitrile-methanol (30:35:35, v/v/v). The following gradient program was used at a flow rate of 200 μL/min: 0–100% B for 20 min, hold 100% B for 10 min, and re-equilibrate to 100% A for 8 min.

## Animal samples

Our animal protocol (HKD43046) was reviewed and approved by the Institutional Animal Care and Use Committee at Hokudo Co., Ltd (Sapporo, Japan). The rat model of CLD has been previously described in detail (*Tamai et al., 2021*). Briefly, Wister male rats (SPF, CLEA Japan: Tokyo, Japan) aged 7 weeks were fed solid normal diet, CE-2 (CLEA Japan), under conventional conditions and were orally administered carbontetrachloride ($CCl_4$) at 1.0 mL/kg twice a week for 4 weeks to induce advanced fibrosis, or cirrhosis, at which point the animals were divided into two groups by weight stratified random sampling. The CLD rats then received daily oral administration of BCAA (10 g/kg/day) (n=10) or 0.9% saline solution (control) (n=10) for 6 weeks. The CLD state was maintained with twice weekly administration of $CCl_4$ at 0.5 mL/kg for 6 weeks (10 weeks total). The rats were individually maintained at a constant temperature (23±3°C), 50±20% relative humidity, and 12 hr light-dark cycles (lights on at 7 am) and had free access to food and water. Analysis of rat number was 9/10 in BCAA group and 8/10 in control group due to death by $CCl_4$ in the experimental term. Wister male rats aged 10 weeks were used as a control, wild-type rats (n=3).

## Gastrocnemius muscle histological analysis and serum BA measurement in rats

All rats were sacrificed at the conclusion of our treatment protocol under anesthesia (isoflurane, DS-pharma, Osaka, Japan). Whole rat blood was collected and allocated into tubes with anticoagulant (EDTA). A portion of gastrocnemius muscle was fixed in 10% formalin for 24 hr and embedded in paraffin, and the remaining gastrocnemius muscle was flash frozen in liquid nitrogen and stored at –80°C. The gastrocnemius muscle sections were prepared and stained for hematoxylin and eosin. All images were taken by Olympus CKX53 (Olympus, Tokyo, Japan) and quantitated using Image J software (NIH Image). Serum BA levels were quantified by LC-MS/MS at CMIC Pharma Science Co., Ltd (Kobe, Japan).

## Cell culture, treatment and, immunofluorescence

C2C12 myoblasts (kindly gift from Dr. Fujita at Tokyo Institute of Technology) were maintained in (Dulbecco's Modified Eagle Medium) DMEM containing 20% fetal bovine serum at 37°C and 5% $CO_2$. The confluent cells were differentiated into myotubes by culturing with DMEM containing 2% horse serum for 5 days with LCA (Millipore-Sigma, Japan), TGR5 agonist (1 μM INT-777, Millipore-Sigma), or TGR5 agonist plus TGR5 antagonist (100 μM SBI-115, Millipore-Sigma). All experiments were repeated twice with three biological replicates in each experiment. For immunofluorescence, cells were fixed with 4% paraformaldehyde for 10 min, permeabilized with 0.5% Triton X-100 for 5 min and then incubated with anti-myosin heavy chain (MHC) antibody (#376157, Santa Cruz, Dallas, TX, USA) at 4°C overnight. MHC and nucleus were visualized with Alexa 488-conjugated anti-mouse antibody and DAPI, respectively. All pictures were taken by KEENC BZ-X710 (KEYENCE, Japan). Changes in cell strength and width were quantified at 5 days post-LCA addition from five fields (total 96 myotubes in control, 109 myotubes in 70 nM LCA and 61 myotubes in 700 nM LCA) using NIH ImageJ software.

## Gene expression

Total RNA was isolated from gastrocnemius muscle or C2C12 cells using TRI Reagent (Molecular Research Center, Cincinnati, OH, USA) according to the manufacturer's instructions. The cDNA was synthesized from total RNA using a cDNA Synthesis kit (Takara, Shiga, Japan). Real-time PCR quantification was performed using the KAPA SYBR FAST quantitative PCR master mix (KAPA Biosystems, Wilmington, MA, USA) or a TaqMan gene expression assay (Thermo Fisher Scientific Inc) for *Sod1* and the 7300 Real-time PCR Detection System (Thermo Fisher Scientific Inc). The PCR primers were used to amplify each gene as listed in *Supplementary file 1*. Mean values of mRNA were normalized to beta 2 microglobulin (*B2m*).

## Western blotting analysis

C2C12 cells were homogenized in Radio-Immunoprecipitation Assay (RIPA) buffer (150 mM NaCl, 1.0% NP-40, 1% sodium deoxycholate, 0.1% sodium dodecyl sulphate, and 50 mM Tris-HCl pH8.0) containing a protease inhibitor cocktail (Millipore-Sigma) and phosphatase inhibitors (Millipore-Sigma). 20 μg of cell lysate was resolved using a TGX gel (Bio-Rad, Hercules, CA, USA), transferred to a polyvinylidenedifluoride membrane, and blotted with the appropriate primary antibody. Membranes were incubated with peroxidase-conjugated secondary antibody (GE Healthcare Bioscience, Marlborough, MA, USA). Protein bands were visualized using an enhanced chemiluminescence reagent (Bio-Rad), digitized using a Lumino-image analyzer (LAS-4000 iniEPUV, Fuji Film, Tokyo, Japan or FUSION SOLO. 7S. EDGE., Vilber, France) and quantitated using the program Multi Gauge (Fuji Film) or Evolution Capt (Vilber). Anti-GAPDH (#60004, Proteintech, Rosemont, IL), anti-phospho-Akt (Ser 473) (#4060, Cell Signaling Technology, Danvers, MA, USA), anti-pan-Akt (#4691, Cell Signaling Technology), anti-phospho-mTOR (Ser 2448) (#5536, Cell Signaling), and anti-mTOR (#2983, Cell Signaling Technology) were used as primary antibodies.

## Statistical analyses

Continuous variables are presented as mean ± SD or median (minimum-maximum), and categorical variables are shown as numbers of patients. The continuous data were compared using the Mann-Whitney U or unpaired t test in two groups or Kruskal-Wallis in multiple groups, and the magnitude of differences was calculated using the effect size analysis (Cohen's d). The relationship between the serum BA levels and clinical data was examined using Spearman's rank correlation coefficient. The categorical data were compared using the Chi-squared test. ROC curves and the corresponding AUC were used to obtain cut-offs for the outcomes. The Youden index was applied to calculate the optimal cut-off point. Overall survival (OS) was measured using the Kaplan-Meier method and compared using the log-rank test. All statistical analyses were performed using SPSS23.0 software (IBM, Armonk, NY, USA) or Prism 9 (GraphPad Software, Inc, CA, USA). Differences were considered to be significant at $p < 0.05$.

**Table 1.** CLD patient baseline clinical and biochemical profiles of CLD patients.

| | n=73 |
|---|---|
| Age, years | 71.0±11.0 |
| Gender, male/female | 58/15 |
| Etiology, HBV/HCV/nonalcoholic steatohepatitis/alcohol/others | 13/21/21/16/2 |
| Barcelona Clinic Liver Cancer (0 /A/B/ C/D) | 11/29/13/19/1 |
| Albumin, g/dL | 4.04±0.49 |
| Total bilirubin, mg/dL | 1.00±0.53 |
| Albumin-bilirubin | −2.65±0.48 |
| Prothrombin time, % | 87.6±18.7 |
| Psoas muscle mass index, $cm^2/m^2$ | 5.13±1.99 |

Data are presented as number of patients, mean ± SD. CLD, chronic liver disease.

The online version of this article includes the following source data for table 1:

**Source data 1.** Serum albumin, total bilirubin, albumin-bilirubin (ALBI), prothrombin time, and psoas muscle mass index (PMI) in chronic liver disease (CLD) patients.

**Table 2.** Baseline bile acids composition.

| | n=73 (mmol/L) |
|---|---|
| Total bile acids | 18.3±17.0 |
| Total of primary bile acids | 12.7±14.0 |
| Cholic acid | 1.30±3.28 |
| Glycocholic acid | 1.71±2.86 |
| Taurocholic acid | 0.35±0.78 |
| Chenodeoxycholic acid | 2.68±4.78 |
| Glycochenodeoxycholic acid | 4.69±5.64 |
| Taurochenodeoxycholic acid | 1.95±4.03 |
| Total of secondary bile acids | 5.58±7.91 |
| Deoxycholic acid | 0.89±1.19 |
| Glycodeoxycholic acid | 0.99±1.88 |
| Taurodeoxycholic acid | 0.19±0.46 |
| Lithocholic acid | 0.067±0.112 |
| Glycolithocholic acid | 0.020±0.048 |
| Taurolithocholic acid | 0.003±0.013 |
| Ursodeoxycholic acid | 1.13±2.38 |
| Glycoursodeoxycholic acid | 2.21±5.56 |
| Tauroursodeoxycholic acid | 0.07±0.24 |

Data are presented as number of patients, mean ± SD.

The online version of this article includes the following source data for table 2:

**Source data 1.** Serum total bile acids (BAs), total primary BAs, total secondary BAs, and BAs composition in chronic liver disease (CLD) patients.

## Results

### Serum LCA levels are positively and significantly correlated with PMI in CLD patients

The clinical features of the 73 (58 men and 15 women) enrolled CLD patients are shown in *Table 1*. The cohort of patients admitted to our study was based on a variety of causative agents: 13 HBV, 21 HCV, 21 NASH, 16 alcoholism, and 2 other factors. Patients infected with HBV or HCV were under infection control, with sustained virological response monitoring, by direct-acting antiviral treatment against HCV or treatment with nucleos(t)ide analogs against HBV in the clinical course of each patient. The Barcelona Clinic Liver Cancer staging showed 11, 29, 13, 19, and 1 patients in Stage 0, A, B, C, and D, respectively.

The patient mean of total serum BAs was 18.3±17.0 µmol/L, composed of primary BAs (12.7±14.0 µmol/L) and secondary BAs (5.58±7.91 µmol/L). The serum level of 15 individual BA compositions is shown in *Table 2*. Total serum primary BA level was negatively correlated with albumin ($r$=−0.456 and p<0.0001) and prothrombin time (PT, %) ($r$=−0.410 and p=0.0003) and was positively correlated with alkaline phosphatase ($r$=0.240 and p=0.011; *Figure 1A*). Furthermore, the total primary BA level was significantly higher in ALBI grades 2 and 3 than in ALBI grade 1 (d=0.74 and p=0.0006; *Figure 1B*). Notably, PMI values were positively and significantly correlated with serum LCA levels ($r$=0.304 and p=0.009) and serum LCA ratio, which was LCA/total BAs ($r$=0.230 and p=0.049), and the only BA composition correlated with PMI (*Figure 1C*). Next, we set out to assess the changes in serum BA composition associated with muscle mass. To do this, the cohort was divided into two groups: low muscle mass, defined by PMI below 6.36 $cm^2/m^2$ for men and 3.92 $cm^2/m^2$ for women (*Hamaguchi et al., 2016*), and normal muscle mass. The level of total primary BAs was decreased, and total secondary BAs were increased in the low muscle mass group compared with the normal muscle mass group (*Figure 1D*). The level of serum TGR5 ligands, CA, CDCA, DCA, and LCA was also decreased in the low muscle mass group (*Figure 1D*). In particular, the level of serum LCA was

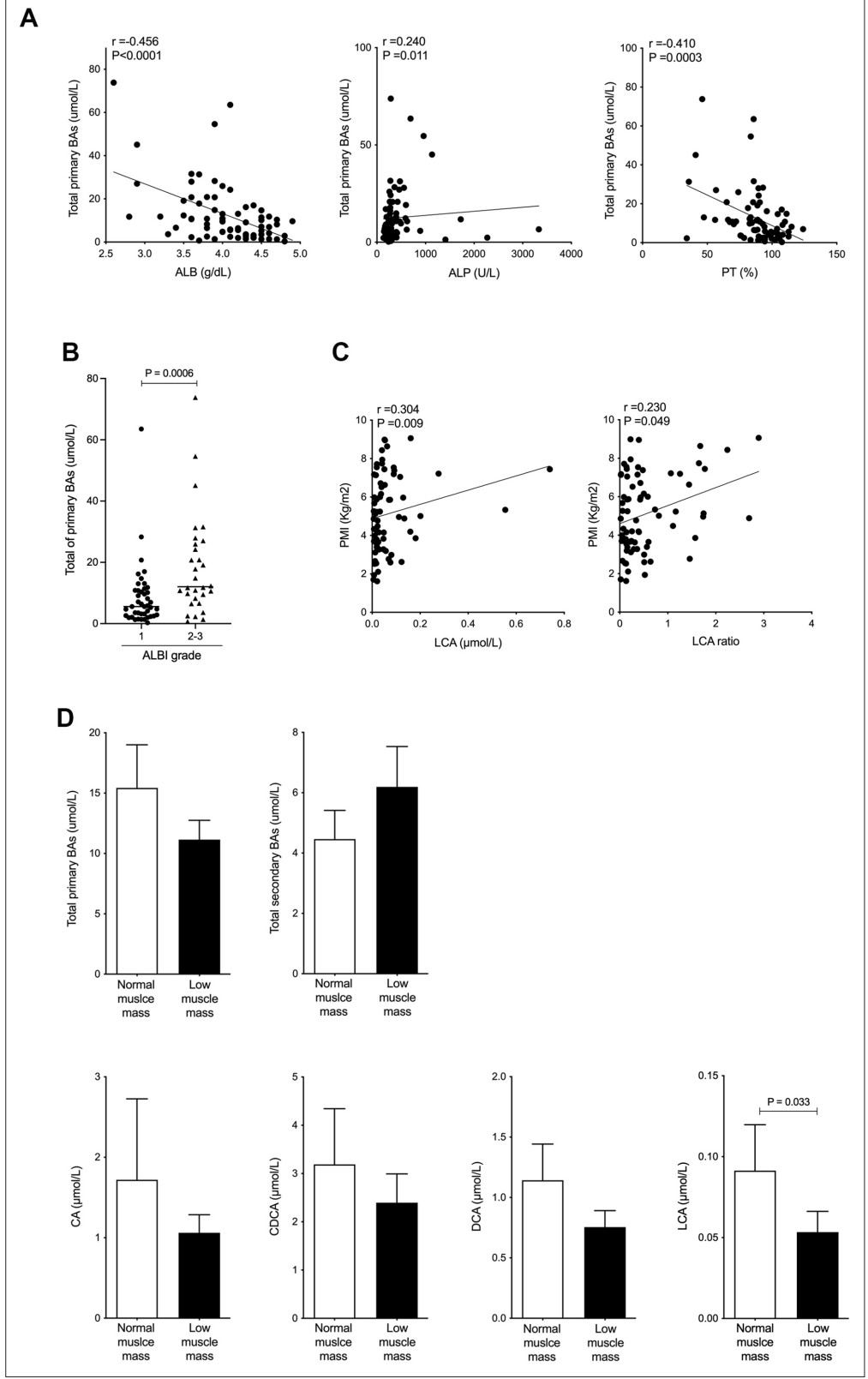

**Figure 1.** Serum LCA level is significantly and positively correlated with PMI in CLD patients and is significantly decreased in CLD patients with low muscle mass. (**A**) Correlation between total primary BAs and ALB, ALP, or PT (%) in CLD patients. (**B**) Changes in total primary BAs in CLD patients with ALBI grade 1 or grades 2–3. (**C**) Correlation of PMI with serum LCA and LCA ratio in CLD patients. (**D**) Changes in total serum primary BAs,

*Figure 1 continued on next page*

*Figure 1 continued*

total secondary BAs, CA, CDCA, DCA, and LCA in CLD patients with normal muscle mass and low muscle mass. Spearman's rank correlation coefficient or Mann-Whitney U test. Values are mean ± SEM. CLD, chronic liver disease; BAs, bile acids; ALBI, albumin-bilirubin; CA, cholic acid; CDCA, chenodeoxycholic acid; DCA, deoxycholic acid; LCA, lithocholic acid; PMI, psoas muscle mass index.

The online version of this article includes the following source data for figure 1:

**Source data 1.** Serum total primary bile acids (BAs), albumin (ALB), alkaline phosphatase (ALP), prothrombin time (PT, %), psoas muscle mass index (PMI), lithocholic acid (LCA), and LCA ratio in chronic liver disease (CLD) patients.

significantly decreased in the low muscle mass group (d=0.34 and p=0.033; *Figure 1D*). These results show that serum LCA levels are indicative of overall muscle mass in CLD patients.

## Serum LCA levels may be a prognostic factor for survival

Finally, we investigated the association between serum BA composition levels and survival. 23 out of 73 patients (31.5 %) died in the average follow-up period of 1005±471 days following our study period. All causes of death were considered HCC related, except one pancreatic cancer and one pneumonia. Serum total primary BA levels were significantly elevated in the deceased (dead) group (d=0.38 and p=0.020), while serum LCA ratio was significantly decreased in the deceased group compared to the survival group (d=0.51 and p=0.042; *Figure 2A*). In other factors, ALBI score was significantly associated with survival (p=0.026), but age, gender, and PT% were not associated with survival. ROC analyses concerning predictors of survival yielded AUC values of 0.670 (95% CI: 0.542–0.797; p=0.021) for total primary BAs and 0.649 (95% CI: 0.519–0.779; p=0.042) for LCA (*Figure 2B*). In the present study, we calculated the ROC analysis survival curve cut-off values for total primary BAs at 10.5 μmol/L (sensitivity 69.6% and specificity 68.0%) and LCA at 0.32 μmol/L (sensitivity 73.9% and specificity 60%). Patients with low total primary BAs (<10.5) showed significantly better OS than patients with high total primary BAs (p=0.0024; *Figure 2C*). Furthermore, patients with high LCA (≥0.32) showed significantly improved OS than patients with low LCA (p=0.0082; *Figure 2C*). These results suggest that serum LCA levels can be useful in predicting patient survival.

## Increased gastrocnemius muscle weight is associated with suppression of protein degradation pathways and elevation of serum LCA levels in CLD rats treated with BCAA

To investigate whether increased gastrocnemius muscle weight is associated with changes in BA composition, we used a CLD rat model administered with $CCl_4$ for 10 weeks (4 weeks to establish advanced fibrosis, or cirrhosis, and an additional 6 weeks to treat with BCAA for attenuation of liver injury), which we have previously reported (*Tamai et al., 2021*). The ratio of gastrocnemius muscle weight to total body weight was significantly increased in CLD rats treated with BCAA (CLD+BCAA) compared to untreated CLD rats (p=0.036; *Figure 3A*). The overall pathological condition of gastrocnemius muscle was similar between CLD and CLD+BCAA rats (*Figure 3B*). Moreover, in concordance with the aforementioned gastrocnemius muscle mass results, gastrocnemius muscle gene expression levels of protein degradation pathways including muscle RING finger 1 (MuRF1), muscle atrophy F-box protein (MafBx), ubiquitin, and E214KDa were notably increased in CLD rats compared with normal rats (indicated as a broken line in the graphs), and MafBx mRNA levels were significantly decreased in gastrocnemius muscle from CLD+BCAA rats (p=0.036; *Figure 3C*). The mRNA levels of the repair gene, transcription factor forkhead box O1 (FOXO1), were notably decreased in CLD rat gastrocnemius muscle samples, but expression recovered in gastrocnemius muscle samples from CLD+BCAA rats (*Figure 3C*). To explore whether increased gastrocnemius muscle mass was associated with BA composition, we measured serum BA levels in CLD and CLD+BCAA rats. Total BAs were dramatically increased in CLD rats when compared to normal rats (indicated as a broken line in a graph) and decreased in CLD+BCAA rats (*Figure 3D*). In line with the total BA data, the ratio of CA, CDCA, and DCA increased in CLD rats and showed a decreasing trend in CLD+BCAA rats (*Figure 3E*).

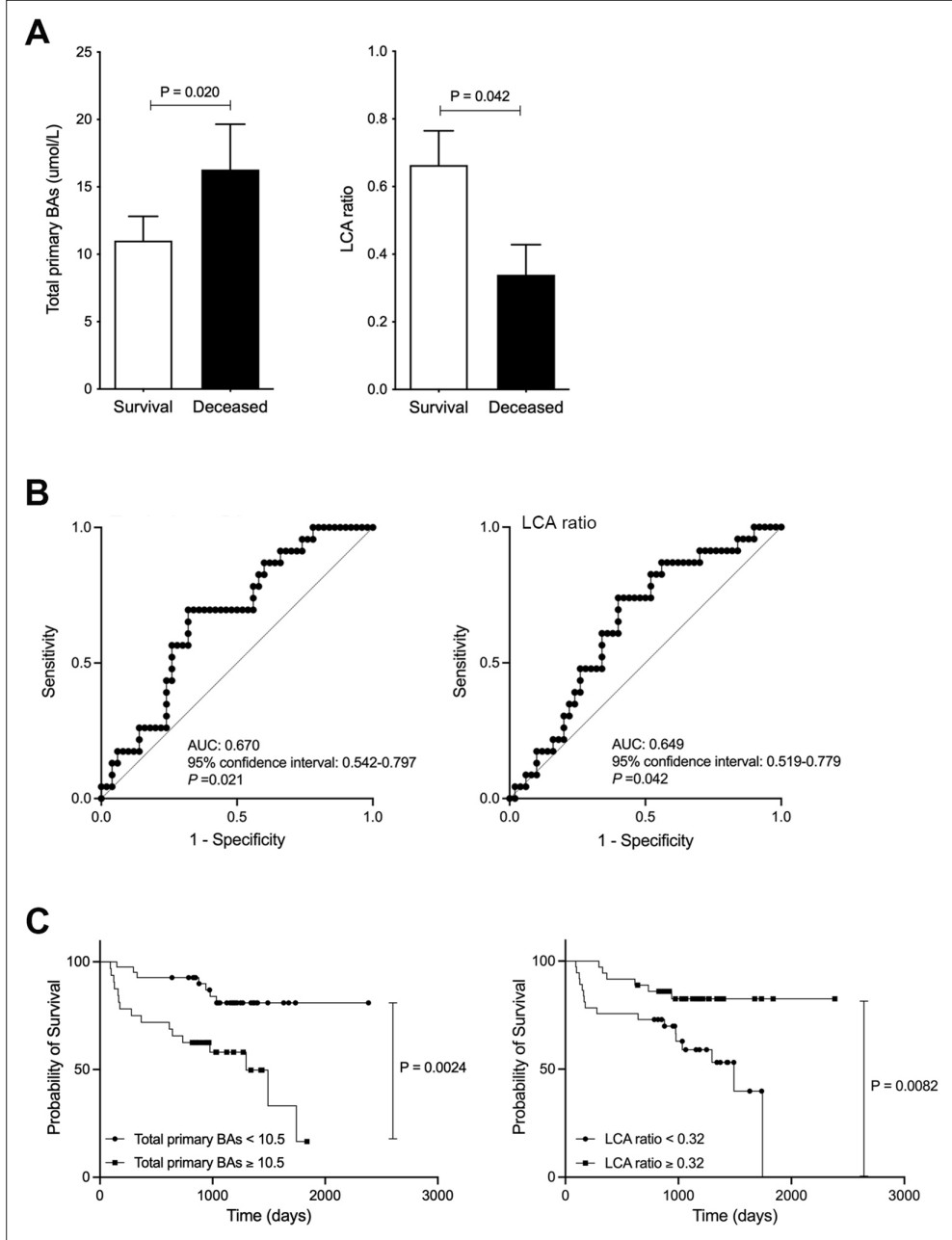

**Figure 2.** Improved survival in chronic liver disease (CLD) patients with high levels of serum lithocholic acid (LCA). (**A**) Serum total of primary bile acids (BAs) and LCA ratio in survival and deceased CLD patients. (**B**) ROC curve of serum total or primary BAs and LCA. (**C**) CLD patient survival curve with total primary BAs and LCA. Correlation of psoas muscle mass index with serum LCA and LCA ratio in CLD patients. Mann-Whitney U test. ROC curves and the corresponding AUC were used to obtain cut-offs for the outcomes. The Youden index was applied to calculate the optimal cut-off point. OS was measured using the Kaplan-Meier method and compared using the log-rank test. Values are mean ± SEM.

The online version of this article includes the following source data for figure 2:

**Source data 1.** Serum total of primary bile acids (BAs) and lithocholic acid (LCA) ratio in survival and deceased chronic liver disease (CLD) patients.

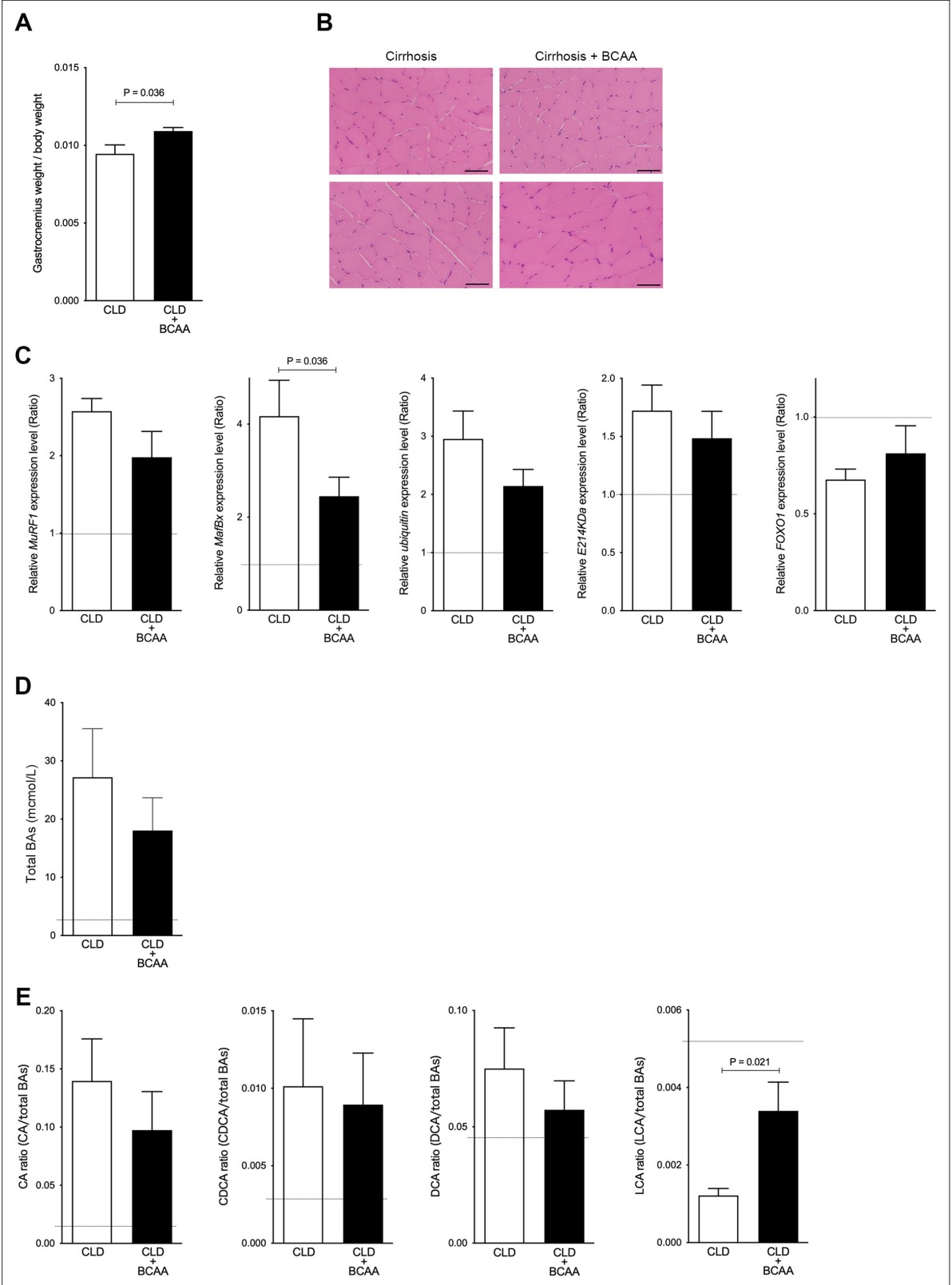

**Figure 3.** Gastrocnemius muscle mass and serum LCA ratio are significantly increased in CLD rats treated with BCAA. (**A**) Changes in gastrocnemius muscle/body weight in CLD rats (n=9) and CLD rats treated with BCAA (CLD+BCAA) (n=8). (**B**) Hematoxylin and eosin staining in gastrocnemius muscle sections from CLD and CLD+BCAA rats. Scale bar, 50 µm. (**C**) Gene expression of *MuRF1*, *MafBx*, *ubiquitin*, *E214KDa*, and *FOXO1* in gastrocnemius muscle from CLD and CLD+BCAA rats as measured by quantitative PCR. All gene expression levels were normalized to housekeeping control, *β2*

*Figure 3 continued on next page*

*Figure 3 continued*

*microglobulin*, and shown relative to the expression levels of control (normal rats). Broken line indicates the expression levels of gastrocnemius muscle from normal rats. (**D and E**) Changes in (**C**) serum total BAs, (**D**) CA/total BAs, CDCA/total BAs, DCA/total BAs, and LCA/total BAs in CLD and CLD+BCAA rats. Broken line indicates the serum BA levels from normal rats. Mann-Whitney U test. Values are mean ± SEM. CLD, chronic liver disease; MuRF1, muscle RING finger 1; MafBx, muscle atrophy F-box protein; FOXO1, forkhead box O1; BAs, bile acids; CA, cholic acid; CDCA, chenodeoxycholic acid; DCA, deoxycholic acid; LCA, lithocholic acid; BCAA, branched-chain amino acid.

The online version of this article includes the following source data for figure 3:

**Source data 1.** Ratio of gastrocnemius weight/body weight in chronic liver disease (CLD) rats and CLD rats treated with branched-chain amino acid (BCAA) (CLD+BCAA).

Notably, the ratio of LCA to total BAs was dramatically decreased in CLD rats and was significantly increased in the CLD+BCAA rat group (p=0.021; *Figure 3E*). These results suggest that an increase in gastrocnemius muscle mass may be associated with serum LCA levels.

## LCA enhances muscle cell hypertrophy through TGR5-IGF-1 pathway

We next examined the effect of LCA on hypertrophy of skeletal muscle cells using C2C12 myoblasts that differentiate rapidly forming myotubes. C2C12 myoblasts were culture for 3 days and approached confluence, then differentiated to myotubes using varying concentrations of LCA (*Figure 4A*). The hypertrophy of C2C12 myotubes was overtly altered in a concentration-dependent manner based on assessment using MHC staining (*Figure 4B*). Corresponding to cell morphological changes, the length and width of the cells were significantly increased in a concentration-dependent manner under quantitative analyses (length, p<0.0001: 0 vs 700 nM and 70 vs 700 nM, p=0.002: 0 and 70 nM; width, p<0.0001 vs 70 or 700 nM and 70 vs 700 nM; *Figure 4C*). Previous reports have shown LCA to be one of the most potent endogenous ligands for TGR5 (*Pols et al., 2011*) capable of inducing IGF-1, which is a known muscle hypertrophy gene (*Sasaki et al., 2018*). In the present study, we found the levels of G-protein-coupled bile acid receptor 1 (*Gpbar1*) mRNA to be significantly increased in C2C12 myotubes treated with LCA (p=0.025: 0 vs 70 nM; *Figure 4D*). Moreover, C2C12 myotubes undergoing LCA-induced hypertrophy showed significantly elevated levels of *Igf1* mRNA (p=0.017: 0 vs 700 nM; *Figure 4D*).

## TGR5 agonist accelerates muscle cell hypertrophy through IGF-1 and Akt activation

To investigate whether LCA-induced TGR5-IGF-1 activation is attenuated by blocking the TGR5 pathway, C2C12 myotubes were coincubated with LCA and TGR5 antagonist (SBI-115). The mRNA levels of *Gpbar1* and *Igf1* were significantly decreased in C2C12 myotubes treated with LCA+SBI-115 compared to LCA alone (p=0.007 and p=0.017, respectively; *Figure 5A*). LCA is the most potent endogenous ligand for TGR5 but also cytotoxic (*Pols et al., 2011*); therefore, LCA alone may not be an appropriate therapeutic target molecule. A TGR5 agonist (INT-777) has been generated and shown to be a useful molecule for TGR5 activation (*Pellicciari et al., 2009*). To explore whether this TGR5 agonist induces muscle cell hypertrophy, differentiated C2C12 myoblasts (myotubes) were incubated with INT-777. INT-777 induced obvious muscle cell hypertrophy when assessed using MHC staining (*Figure 5B*). INT-777 also elevated the mRNA levels of *Gpbar1* and *Igf1* (p=0.003 and p=0.0004, respectively; *Figure 5C*). IGF-1 is known to activate the PI3K-Akt pathway, thus leading to stimulation of protein synthesis, resulting in accelerated muscle hypertrophy (*Sartori et al., 2021*). Indeed, the ratio of Akt phosphorylation against to total Akt was significantly increased in C2C12 myotubes treated with INT-777 (p=0.0006; *Figure 5D*). As one of downstream proteins, the ratio of mTOR phosphorylation against to total mTOR was also increased in C2C12 myotubes treated with INT-777, although it was not significant (*Figure 5D*). These results suggest that the TGR5-IGF-1-Akt pathway has an important role in muscle hypertrophy.

## Discussion

In the present study, we demonstrated that serum LCA levels and LCA ratio were positively associated with skeletal muscle mass in CLD rats treated with BCAA, as well as human subjects, and that LCA-induced skeletal muscle cell hypertrophy occurs through TGR5-IGF-1-Akt3 activation. BCAA

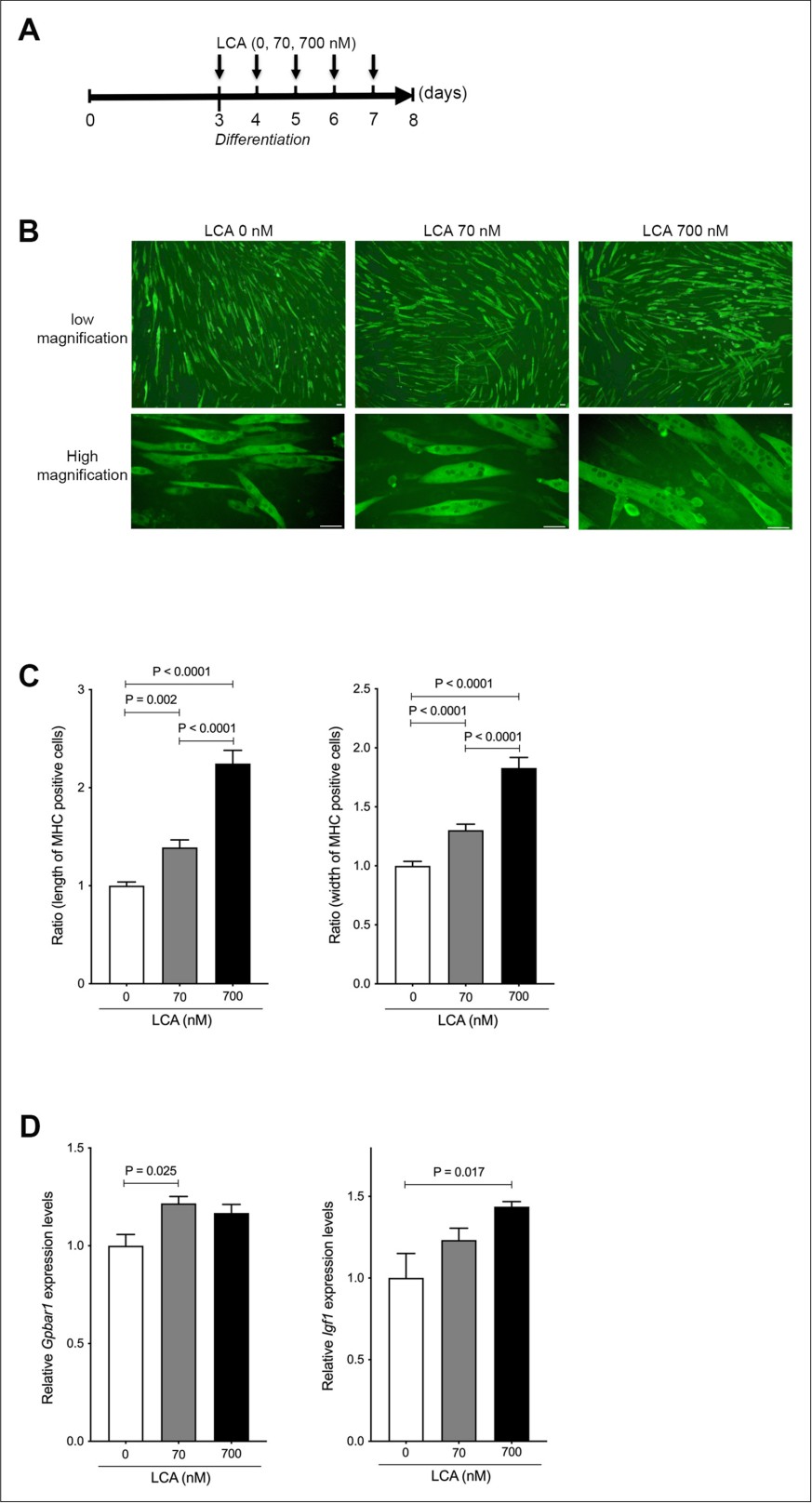

**Figure 4.** LCA induces hypertrophy of skeletal muscle cells. (**A**) Scheme of experimental design in C2C12 myoblast to myotubes treated with LCA. (**B**) Myosin heavy chain (MHC) staining in C2C12 myotubes treated with 0, 70, and 700 nM of LCA. Scale bar, 50 µm. (**C**) Changes in length and width of MHC positive cells quantified from *Figure 2B*. (**D**) Gene expression of *Gpbar1* and *Igf1* in C2C12 myotubes treated with 0, 70, and 700 nM of LCA. Kruskal-Wallis

*Figure 4 continued on next page*

*Figure 4 continued*

test. Values are mean ± SEM from three biological replicates. LCA, lithocholic acid; Gpbar1, G-protein-coupled bile acid receptor 1; IGF, insulin growth factor.

The online version of this article includes the following source data for figure 4:

**Source data 1.** Ratio of length and width of myosin heavy chain (MHC) positive cells quantified from MHC staining in C2C12 myotubes treated with 0, 70, and 700 nM of lithocholic acid (LCA).

supplementation is approved for use in CLD patients within the clinical setting as a means to provide compensatory albumin, thus maintaining liver function (*European Association for the Study of the Liver. Electronic address: easloffice@easloffice.eu and European Association for the Study of the Liver, 2019*), as well as increased muscle mass associated with an acceleration of the taurocholic acid cycle (*Ismaiel et al., 2022*). In our previous study, we reported that hepatocellular damage was attenuated using BCAA supplementation as a result of improved lipid metabolism and mitochondrial damage repair in CLD rats (*Tamai et al., 2021*). Using the same CLD rat model in the current study, we revealed that gastrocnemius muscle mass was significantly increased using BCAA treatment. BCAA treatment has direct effects on liver and skeletal muscle; however, we hypothesized that one or more CLD-related molecules/factors might regulate skeletal muscle mass via a liver-muscle axis. Indeed, we found that the serum LCA ratio (LCA/total BAs) was significantly increased in CLD rats, which also showed an increase in gastrocnemius muscle mass. Furthermore, we showed that serum LCA and LCA ratio were significantly and positively associated with PMI in CLD patients. These results from CLD rats and human subjects suggest that LCA can regulate muscle mass via a liver-muscle axis, although further studies are warranted in the future using a greater number of patients as part of a multicenter study.

The role of LCA in the progression of CLD has not been fully developed due to the lack of general sensitivity in the system used to measure BA composition and is therefore not sufficient to detect low levels of LCA in the blood. Our established highly sensitive system for BA composition (*Murakami et al., 2018*) allows us to detect all aspects of BA composition resulting in the discovery of a new role for LCA, which is a positive correlation of serum BA composition with skeletal muscle mass in CLD patients. Furthermore, we revealed that a decrease in serum LCA level portends a worse survival outcome in CLD patients with associated low muscle mass. CLD patients with sarcopenia, defined by low muscle mass and power, also display decreased survival when compared to CLD patients without sarcopenia (*Hara et al., 2016*), thus serum LCA may be a useful measure to monitor sarcopenia in CLD patients. Current reports have also demonstrated that LCA is one of the most potent anti-bacterial agents, selective against gram-positive bacteria, resulting in a longer lifespan of centenarians (*Sato et al., 2021*), and is one of the most potent endogenous ligands for TGR5, which protects against alcohol-induced liver steatosis and inflammation in mice (*Iracheta-Vellve et al., 2018*). This evidence clearly shows that LCA plays a critical role in the progression of CLD and intestinal microbiota as a function of the liver-gut axis. Our latest results presented here, associating LCA with skeletal muscle mass, will lead to new insights into the role of LCA as a component of the liver-muscle-gut axis.

In CLD rats and human subjects, we observed an association between gastrocnemius muscle mass and LCA only, although serum CA, CDCA, and DCA levels also showed a decreasing trend in CLD patients with low muscle mass. This result is reasonable since the hierarchy of BA affinity for TGR5 is as follows: LCA>DCA> CDCA>CA (*Sato et al., 2007*). We also demonstrated that a TGR5 antagonist induced skeletal muscle cell hypertrophy through IGF-1 activation, but we need further studies to develop a new antagonist with similar affinity of LCA to the TGR5 binding pocket minus the cytotoxicity aspect.

In conclusion, we revealed new roles for LCA as a positive regulator of skeletal muscle mass in both CLD rats and human patients and as a mediator of skeletal muscle cell hypertrophy in differentiated C2C12 myoblasts (myotubes). The serum LCA ratio measurement was significantly decreased in CLD patients with low muscle mass. Current results suggest that serum LCA levels

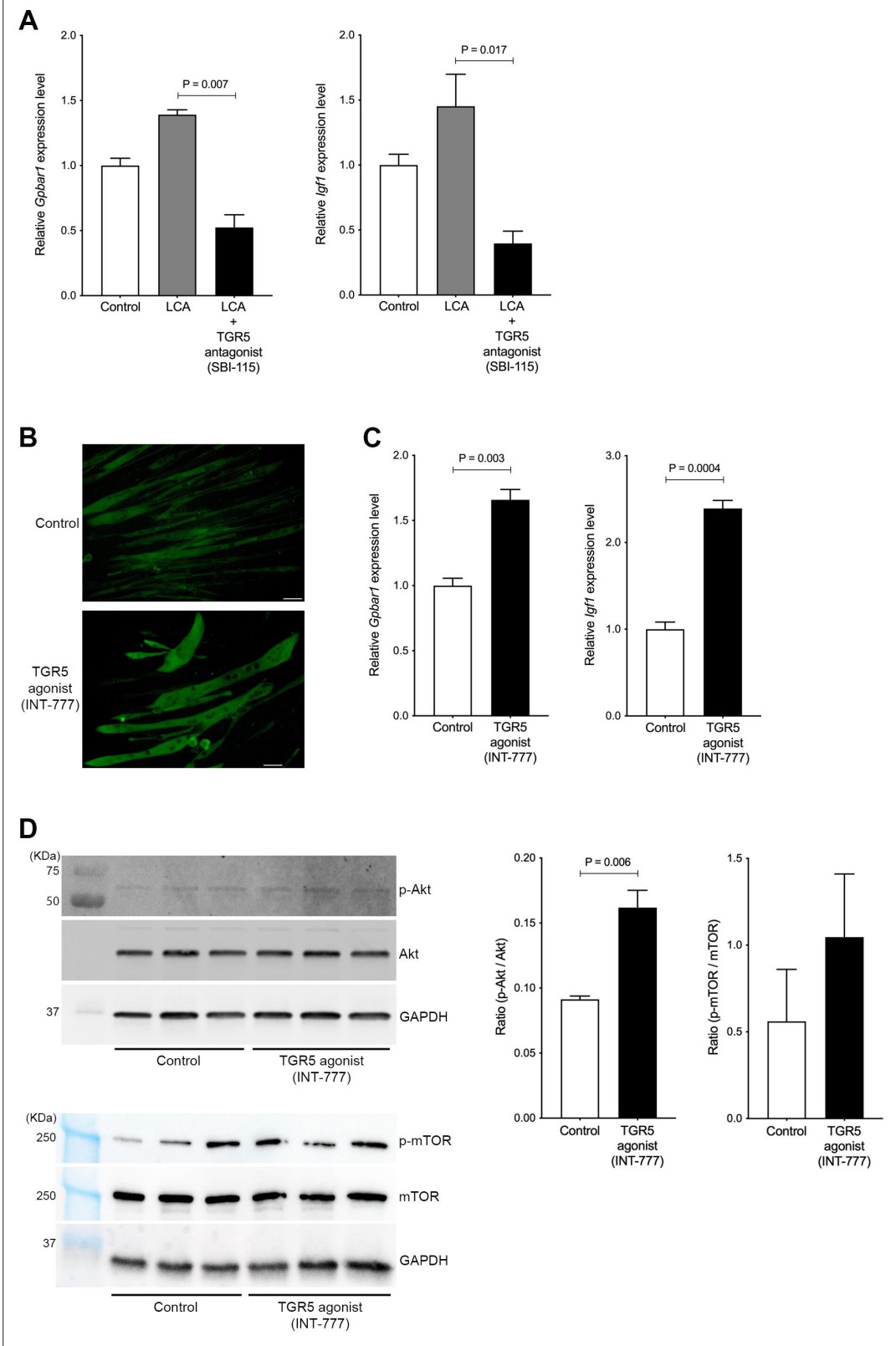

**Figure 5.** Hypertrophy of skeletal muscle cells is induced by TGR5-IGF-1-Akt3 activation. (**A**) Gene expression of *Gpbar1* and *Igf1* in C2C12 myotubes treated with 70 nM LCA and 70 nM LCA plus 100 µM of TGR5 antagonist, SBI-115. (**B**) Myosin heavy chain staining in C2C12 myotubes treated with 1 µM of TGR5 agonist, INT-777. Scale bar, 50 µm. (**C**) Gene expression of *Gpbar1* and *Igf1* in C2C12 myotubes treated with 1 µM of INT-777.

*Figure 5 continued on next page*

*Figure 5 continued*

(**D**) Protein expression of phosphorylated Akt (p-Akt), Akt, phosphorylated mTOR (p-mTOR), mammalian target of rapamycin (mTOR), and GAPDH measured by western blotting in C2C12 myotubes treated with 1 µM of INT-777. Quantification of p-Akt/Akt or p-mTOR/mTOR from western blotting membrane. Kruskal-Wallis test or unpaired t test. Values are mean ± SEM from three biological replicates. LCA, lithocholic acid; Gpbar1, G-protein-coupled bile acid receptor 1; IGF, insulin growth factor; Akt, AKT serine/threonine kinase; GAPDH, glyceraldehyde-3-phosphate dehydrogenase.

The online version of this article includes the following source data for figure 5:

**Source data 1.** The uncropped blots with the relevant bands.

**Source data 2.** The original blots for Akt.

**Source data 3.** The original blots for phosphorylated Akt (p-Akt).

**Source data 4.** The original blots for glyceraldehyde-3-phosphate dehydrogenase (GAPDH) (for Akt).

**Source data 5.** The original blots for mammalian target of rapamycin (mTOR).

**Source data 6.** The original blots for phosphorylated mammalian target of rapamycin (p-mTOR).

**Source data 7.** The original blots for glyceraldehyde-3-phosphate dehydrogenase (GAPDH) (for mammalian target of rapamycin [mTOR]).

may be used as a prognostic factor of survival in CLD patients with sarcopenia, and a TGR5 agonist holds the potential to be a candidate as a therapeutic target in the prevention of sarcopenia in CLD patients.

## Acknowledgements

We would like to thank Dr. Teruo Miyazaki, Dr. Akira Honda, and Dr. Tadashi Ikegami in Department of Gastroenterology, Tokyo Medical University Ibaraki medical center for measurement of human BAs.

## Additional information

### Funding

| Funder | Grant reference number | Author |
|---|---|---|
| Japan Society for the Promotion of Science | 22K08011 | Motoh Iwasa |
| Japan Society for the Promotion of Science | 21H02892 | Hayato Nakagawa |
| Japan Agency for Medical Research and Development | JP21fk0210090 | Hayato Nakagawa |
| Japan Agency for Medical Research and Development | JP22fk0210115 | Hayato Nakagawa |
| Otsuka Pharmaceutical | | Motoh Iwasa |

The funders had no role in study design, data collection and interpretation, or the decision to submit the work for publication.

### Author contributions

Yasuyuki Tamai, Data curation, Writing – original draft; Akiko Eguchi, Conceptualization, Supervision, Methodology, Writing – original draft, Project administration; Ryuta Shigefuku, Mina Tempaku, Ryosuke Sugimoto, Motoh Iwasa, Data curation, Writing - review and editing; Hiroshi Kitamura, Investigation, Methodology, Writing - review and editing; Yoshinao Kobayashi, Formal analysis, Investigation, Writing - review and editing; Yoshiyuki Takei, Writing - review and editing; Hayato Nakagawa, Investigation, Writing - review and editing

## Author ORCIDs

Yasuyuki Tamai ![ORCID] http://orcid.org/0000-0002-8012-8578
Akiko Eguchi ![ORCID] http://orcid.org/0000-0002-0555-2707
Yoshinao Kobayashi ![ORCID] http://orcid.org/0000-0003-2447-684X

## Ethics

Human subjects: The study protocol (H2019-063) was approved by the Clinical Research Ethics Review Committee of Mie University Hospital. This study was performed retrospectively on stored samples, and subjects were allowed to opt out of their data being used. Written informed consent was obtained from all subjects at the time of blood sampling.

Our animal protocol (HKD43046) was reviewed and approved by the Institutional Animal Care and Use Committee at Hokudo Co., Ltd (Sapporo, Japan).

## Decision letter and Author response

Decision letter https://doi.org/10.7554/eLife.80638.sa1
Author response https://doi.org/10.7554/eLife.80638.sa2

---

## Additional files

### Supplementary files

• MDAR checklist
• Supplementary file 1. Primer list for quantative PCR.

### Data availability

All data generated or analysed during this study are included in the manuscript and supporting file.

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
