## [Editor Report]

Alterations in skeletal muscle mass, in particular sarcopenia, are of central clinical importance. This paper examines a particular subclass of etiologies related to the chronic liver disease for this condition. This paper identifies a potential causative for this for the first time by using both in vivo and in vitro techniques to draw associations between bile acid concentrations and liver disease. The paper is of interest to both basic scientists and clinicians.

---

## [Decision Letter]

**Decision letter after peer review:**

Thank you for submitting your article "Association of lithocholic acid with skeletal muscle hypertrophy through TGR5-IGF-1 and skeletal muscle mass in chronic liver disease rats and humans" for consideration by *eLife*. Your article has been reviewed by 3 peer reviewers, one of whom is a member of our Board of Reviewing Editors, and the evaluation has been overseen by Mone Zaidi as the Senior Editor. The reviewers have opted to remain anonymous.

Essential revisions:

Major comments.

Reviewer 1 was positively impressed by the subject area and the detailed nature of the analysis but did not offer a detailed expert report.

Reviewer 2 had the following major comments to make that require addressing, relating to each of the broad issues in the paper:

(A) Tamai et al. aimed to investigate the association between bile acid concentration (lithocholic acid) in both humans and an in vivo rat model of chronic liver disease induced by CCl4 treatment. They show a decline in bile acid concentrations in CLD rats treated with BCAA, and an increase in concentrations in CLD rats, without BCAA treatment, compared to controls. This decline in LCA concentration was negatively associated with gastrocnemius muscle mass (i.e., muscle mass increased).

However, lithocholic acid was associated with an increase in gastrocnemius muscle mass in the CLD rats + BCAA group. The authors also highlight a significant, positive correlation between psoas muscle index in CLD patients and lithocholic acid concentrations.

(B) The authors also highlight a significant, positive correlation between psoas muscle index in CLD patients and lithocholic acid concentrations. Of interest, the authors treated C2C12 skeletal muscle cells, an immortalised rat cell line with lithocholic acid in order to investigate a mechanistic link between the associations observed within in vivo data. The authors suggest that the TGF5-IGF-1-Akt3 pathway may be linked to muscle hypertrophy.

However, these suggestions are not fully supported by the data within the manuscript and need to be extended further to downstream targets related to mTOR, and ultimately muscle protein synthesis.

(C) The multi-model approach is a clear strength of this manuscript. Both animal and human work were used to investigate associations in LCA concentration and muscle mass. The authors also use an in vitro model to investigate potential mechanistic links which may explain the associations identified within in vivo models.

However, the manuscript does have a number of flaws. Firstly, the authors do not set out specific and detailed aims in relation to their in vitro work. Secondly, the rationale for the inclusion of BCAA treatment in rats does not come across in the manuscript.

(D) Additionally, the flow throughout the manuscript, in particular, the Results section needs improvement.

Reviewer 3 had the following major comment to make:

Introduction

1. Page 4, line 66.

It has been reported that a decrease in vitamin D concentration is also associated with the cause of sarcopenia associated with liver disease. If possible, please describe this point and describe the reference (Okubo T et al., Hepatology Research 2020. PMID: 31914479).

Reviewer 2 had the following specific comments to make:

(A) Structure of the manuscript: There is a significant cause for concern in relation to the structure of the manuscript. Firstly, there are no hypotheses mentioned in the introduction, nor aims that support the inclusion of the in vitro listed. This manuscript should be proofread and undergo major revisions prior to any acceptance. Additionally, the authors make bold statements that are not supported by the data in this manuscript in relation to the in vitro data. They may wish to tone down the wording or remove certain statements to highlight certain associations.

(B) Detailed queries:

Comment 1: The title indicates that only rats and humans are used in this work. Do the authors also use an in vitro model? Could the title be altered the reflect the full multi-approach used here?

Comment 2: Page 4 line 62. Sarcopenia is the loss of muscle mass, strength, and physical function. The difference between 'hepatic' vs. 'aging' sarcopenia is in the definition of primary and secondary sarcopenia, with primary being age-related and secondary encompassing other causes such as chronic disease. Please amend.

Comment 3: Page 4 line 68. IGF-1 – mTOR pathway?

Comment 4: Page 4 line 72. Skeletal muscle mass index?

Comment 5: Page 5 lines 80-85. Please can the authors provide references for this passage of text?

Comment 6: Page 6: Can the authors please set out their aims to encompass all included work? At present there is no mention of C2C12s in this section, thus it is unclear what the point of their inclusion is without deducing further along.

Comment 7: Where appropriate, please can the authors provide actual p-values as opposed to p< 0.05, etc. Additionally, the inclusion of effect size would be a valuable inclusion.

Comment 8: The Results section should be restructured. As a reader, it would make more sense to begin with human data to show the association with muscle mass in patients, prior to in vivo rat models and in vitro models. This is due to the animal work containing an additional arm (BCAA) to investigate the relationship between LCA and an increased muscle mass, in addition to a decline in muscle mass, which is only investigated in humans. Furthermore, this would follow nicely onto the C2C12 mechanistic data.

Comment 9: Page 18 Line 337. Can the authors please clarify what they are referring to when measuring 'cell strength'? Additionally, please can the authors clarify how width was quantified? If multiple measures were not taken across each myotube, and a significant number of fields of view imaged for analysis results will not be accurate.

Comment 10: Please can the authors provide a rationale for their inclusion of limited gene and protein analysis. It is unclear why the Akt gene expression was not expressed, nor other key markers at the protein level. The data in this manuscript does not support the conclusions drawn. An increase in MPS cannot be concluded from the investigation of only IGF and Akt. Other downstream proteins are required to reach such a conclusion with more support, most notably mTOR (Page 9 lines 159-161).

Reviewer 3 had the following specific comments to make:

Results (Page 11, line 194)

1. The authors described that 23 of 73 patients died. Were all the causes of death in these patients HCCs?

2. Regarding the paragraph and Figure 5, what is decreased group? 'Decreased' means muscle volume decreased? I think this point is difficult for readers to understand. Please describe this paragraph and Figure 5 including the legend a little more clearly (e.g. …LCA levels were decreased in the decreased group…).

3. The authors have shown that BAs are associated with survival, but was this confounded with other factors? So, were BAs an independent and significant factor related to survival?

*Reviewer #1 (Recommendations for the authors):*

My wishing that this paper proceeds to review was based on my positive impression of this paper and the importance of the topic, but I shall defer to the detailed expertise of the reviewers in their reports.

*Reviewer #2 (Recommendations for the authors):*

There is a significant cause for concern in relation to the structure of the manuscript. Firstly, there are no hypotheses mentioned in the introduction, nor aims which support the inclusion of the in vitro listed. This manuscript should be proofread and undergo major revisions prior to any acceptance. Additionally, the authors make bold statements that are not supported by the data in this manuscript in relation to the in vitro data. They may wish to tone down the wording or remove certain statements to highlight certain associations.

Comment 1: The title indicates that only rats and humans are used in this work. Do the authors also use an in vitro model? Could the title be altered the reflect the full multi-approach used here?

Comment 2: Page 4 line 62. Sarcopenia is the loss of muscle mass, strength, and physical function. The difference between 'hepatic' vs. 'aging' sarcopenia is in the definition of primary and secondary sarcopenia, with primary being age-related and secondary encompassing other causes such as chronic disease. Please amend.

Comment 3: Page 4 line 68. IGF-1 – mTOR pathway?

Comment 4: Page 4 line 72. Skeletal muscle mass index?

Comment 5: Page 5 lines 80-85. Please can the authors provide references for this passage of text?

Comment 6: Page 6: Can the authors please set out their aims to encompass all included work? At present there is no mention of C2C12s in this section, thus it is unclear what the point of their inclusion is without deducing further along.

Comment 7: Where appropriate, please can the authors provide actual p-values as opposed to p< 0.05, etc. Additionally, the inclusion of effect size would be a valuable inclusion.

Comment 8: The Results section should be restructured. As a reader, it would make more sense to begin with human data to show the association with muscle mass in patients, prior to in vivo rat models and in vitro models. This is due to the animal work containing an additional arm (BCAA) to investigate the relationship between LCA and an increased muscle mass, in addition to a decline in muscle mass, which is only investigated in humans. Furthermore, this would follow nicely onto the C2C12 mechanistic data.

Comment 9: Page 18 Line 337. Can the authors please clarify what they are referring to when measuring 'cell strength'? Additionally, please can the authors clarify how width was quantified? If multiple measures were not taken across each myotube, and a significant number of fields of view imaged for analysis results will not be accurate.

Comment 10: Please can the authors provide a rationale for their inclusion of limited gene and protein analysis. It is unclear why Akt gene expression was not expressed, nor other key markers at the protein level. The data in this manuscript does not support the conclusions drawn. An increase in MPS cannot be concluded from the investigation of only IGF and Akt. Other downstream proteins are required to reach such a conclusion with more support, most notably mTOR (Page 9 lines 159-161).

---

## [Author Response]

Reviewer #2 (Recommendations for the authors):There is a significant cause for concern in relation to the structure of the manuscript. Firstly, there are no hypotheses mentioned in the introduction, nor aims which support the inclusion of the in vitro listed. This manuscript should be proofread and undergo major revisions prior to any acceptance. Additionally, the authors make bold statements that are not supported by the data in this manuscript in relation to the in vitro data. They may wish to tone down the wording or remove certain statements to highlight certain associations.Comment 1: The title indicates that only rats and humans are used in this work. Do the authors also use an in vitro model? Could the title be altered the reflect the full multi-approach used here?

Thank you for your comment. We now changed the title “Association of lithocholic acid with skeletal muscle hypertrophy through TGR5-IGF-1 and skeletal muscle mass in cultured mouse myotubes, chronic liver disease rats and humans.”

Comment 2: Page 4 line 62. Sarcopenia is the loss of muscle mass, strength, and physical function. The difference between 'hepatic' vs. 'aging' sarcopenia is in the definition of primary and secondary sarcopenia, with primary being age-related and secondary encompassing other causes such as chronic disease. Please amend.

Thank you for your comments. We now amended the definition of primary and secondary sarcopenia “Sarcopenia is the loss of muscle mass, strength and physical function. Sarcopenia is classified into two categories, primary being age-related and secondary encompassing other causes such as chronic diseases.” (Page 4, line 67-69)

Comment 3: Page 4 line 68. IGF-1 – mTOR pathway?

We changed to IGF-1-mTOR signaling (Page 4, line 76-Page 5, line 77).

Comment 4: Page 4 line 72. Skeletal muscle mass index?

We appreciate your comment. We now changed to skeletal muscle mass index (Page 5, line 80).

Comment 5: Page 5 lines 80-85. Please can the authors provide references for this passage of text?

We added references (Arab et al., 2017 and Guzior and Quinn, 2021).

Comment 6: Page 6: Can the authors please set out their aims to encompass all included work? At present there is no mention of C2C12s in this section, thus it is unclear what the point of their inclusion is without deducing further along.

We agreed with your comment and added the words“ in cultured mouse myotubes, C2C12”.

“In this study, we investigate the interaction between BAs, including LCA, and skeletal muscle mass, in CLD patients, as well as CLD rats, and explore the molecular mechanism of LCA on skeletal muscle hypertrophy in cultured mouse myotubes, C2C12” (Page 6, line 105-107).

Comment 7: Where appropriate, please can the authors provide actual p-values as opposed to p< 0.05, etc. Additionally, the inclusion of effect size would be a valuable inclusion.

We provided actual p-values. We also calculated Cohen’s d in CLD patients and added d values in the Result sections (Page 13, line 244 and Page 14, line 255, 263 and 264).

Comment 8: The Results section should be restructured. As a reader, it would make more sense to begin with human data to show the association with muscle mass in patients, prior to in vivo rat models and in vitro models. This is due to the animal work containing an additional arm (BCAA) to investigate the relationship between LCA and an increased muscle mass, in addition to a decline in muscle mass, which is only investigated in humans. Furthermore, this would follow nicely onto the C2C12 mechanistic data.

We greatly appreciate your suggestion. We agreed and restructured the abstract, result and method sections from human, rats and C2C12 experiments.

Comment 9: Page 18 Line 337. Can the authors please clarify what they are referring to when measuring 'cell strength'? Additionally, please can the authors clarify how width was quantified? If multiple measures were not taken across each myotube, and a significant number of fields of view imaged for analysis results will not be accurate.

We measured cell strength at 5 days post-LCA addition. We took five images (fields) and measured length and width in each myotubes (total 96 myotubes in control, 109 myotubes in 70 nM LCA and 61 myotubes in 700 nM LCA) as shown in Figure 4-source data.

We now added detail information in the Method sections. “Changes in cell strength and width were quantified at 5 d post-LCA addition from five fields (total 96 myotubes in control, 109 myotubes in 70 nM LCA and 61 myotubes in 700 nM LCA) using NIH ImageJ software.” (Page 10, line 184-186)

Comment 10: Please can the authors provide a rationale for their inclusion of limited gene and protein analysis. It is unclear why Akt gene expression was not expressed, nor other key markers at the protein level. The data in this manuscript does not support the conclusions drawn. An increase in MPS cannot be concluded from the investigation of only IGF and Akt. Other downstream proteins are required to reach such a conclusion with more support, most notably mTOR (Page 9 lines 159-161).

We performed new experiment to address mTOR activation by western blotting and found that the ratio of mTOR phosphorylation against to total mTOR was increased in C2C12 myotubes treated with INT-777 as a TGR5 agonist (Figure 5D). This result suggest that the downstream protein is also activated through TGR5-IGF-1-Akt pathway.

We now added this result and changed words for tone down the conclusion in the Results section “As one of downstream proteins, the ratio of mTOR phosphorylation against to total mTOR was also increased in C2C12 myotubes treated with INT-777, although it was not significant (Figure 5D). These results suggest that the TGR5-IGF-1-Akt pathway have an important role in muscle hypertrophy. (Page 18-19, line 340-343)

Reviewer 3Introduction1. Page 4, line 66.It has been reported that a decrease in vitamin D concentration is also associated with the cause of sarcopenia associated with liver disease. If possible, please describe this point and describe the reference (Okubo T et al. Hepatology Research 2020. PMID: 31914479).

Thank you for your comments. We now added “decreased vitamin D levels” with reference in the introduction sections (Page 4, line 75).